# Flame-Retardant Thermoplastic Polyether Ester/Aluminum Butylmethylphosphinate/Phenolphthalein Composites with Enhanced Mechanical Properties and Antidripping

**DOI:** 10.3390/polym16040552

**Published:** 2024-02-18

**Authors:** Xue Yang, Yan Zhang, Jia Chen, Liyong Zou, Xuesong Xing, Kangran Zhang, Jiyan Liu, Xueqing Liu

**Affiliations:** 1Institute of Noise & Vibration, Naval University of Engineering, Wuhan 430033, China; 2School of Polymer Materials and Engineering, Jianghan University, Wuhan 430056, China; 3Key Laboratory of Optoelectronic Chemical Materials and Devices of Ministry of Education, Jianghan University, Wuhan 430056, China

**Keywords:** thermoplastic polyether-ester, flame retardancy, metal phosphinate, phenolphthalein, mechanical properties

## Abstract

Aluminum butylmethylphosphinate AiBMP as a flame retardant and phenolphthalein as a synergistic agent were applied in a thermoplastic polyester elastomer (TPEE)) in the current study. The thermal properties, flame retardancy, crystallization and mechanical properties of TPEE/AiMBP with or without phenolphthalein were investigated using various characterizations, including the limiting oxygen index (LOI), vertical burning test (UL 94), thermogravimetric analysis TG, differential scanning calorimetry, microcombustion calorimeter (MCC), scanning electron microscopy (SEM), and mechanical tests. The results revealed that AiBMP alone is an efficient flame retardant of TPEE. Adding 15 wt.% AiBMP increases the LOI value of TPEE from 20% to 36%. The formula TPEE-15 AiBMP passed the UL 94 V-0 rating with no dripping occurring. The MCC test shows that AiBMP depresses the heat release of TPEE. In comparison with pure TPEE, the heat release rate at peak temperature and the heat release capacity of TPEE-15AiBMP are reduced by 46.1% and 55.5%, respectively. With the phenolphthalein added, the formula TPEE/13AiBMP/2Ph shows a higher char yield at high temperatures (>600 °C), and the char layer is stronger and more condensed than TPEE-15AiBMP.The tensile strength and elongation at break values of TPEE-13AiBMP-2Ph are increased by 29.63% and 4.8% in comparison with TPEE-15AiBMP. The SEM morphology of the fracture surface of the sample shows that phenolphthalein acts as a plasticizer to improve the dispersion of AiBMP within the matrix. The good char charming ability of phenolphthalein itself and improved dispersion of AiBMP make the TPEE composites achieve both satisfying flame retardancy and high mechanical properties.

## 1. Introduction

Thermoplastic polyether-ester elastomer (TPEE) is produced by combining crystalline (hard) and amorphous (soft) segments to offer properties of a thermal set elastomer with the processability and recycling ability of thermoplastics [1]. They exhibit exceptional toughness, impact resistance, load-bearing capacity, and low-temperature flexibility and are widely used in electronics and electrical appliances, communications, and automotive industries [2,3]. TPEE has inherent flammability and serious dripping during combustion and involves a real hazard to the users of these applications. Therefore, flame retardancy treatment is a requirement [4,5].

Adding flame retardant additives via physical blending is an easy and economical approach to improve the fire resistance of TPEE. Many of the commercially available systems for the flame retardancy of TPEE consist of a halogen-containing additive and a synergistic agent [6]. With the strict limitation on hazardous substances released into the environment during the disposal of electrical and election waste, the industries for these halogen additive-based TPEEs are under pressure to change to flame retardants that are more environmentally friendly and harmless to health [7].

Phosphorus-based flame retardants have become a hot research topic due to their eco-friendly behaviors and efficient flame-retardant properties [8]. In addition, it has been proved that metal components can efficiently enhance the flame retardancy of polymer composites. Metal components bonded with a flame retardant’s chemical nature can strengthen the thermal properties and flammability of polymers [9]. Moreover, different compositions of metal components and the chemical nature of flame retardants demonstrated that self-reinforced composite properties can be modified to achieve better properties [10].

The metal salt of phosphinates, with high phosphorus content, good thermal stability, and low affinity to moisture, has been developed as an eco-friendly flame retardant in recent years [11]. Metal phosphinates have advantages such as low water absorption, low current leakage, and high thermal stability. They have received great attention in flame-retardant material for the electrical and electronics (E&E) industry [6,7]. The flame-retardant effectiveness of the metal salt of phosphinates is related to the applied polymer and is also related to its chemical structure [12,13]. Commercial aluminum or zinc diethylphosphinate (AlPi or ZnPi) was an effective flame retardant for polyesters [14,15,16,17,18] and polyamide [19] epoxy resin [20,21]. However, AlPi or ZnPi alone in TPEE cannot achieve satisfying performance on fire retardancy at a dosage below 20 wt%. In addition, the mechanical properties of TPEE are seriously damaged at the loading of the filler to meet the flame retardancy requirement [22].

Balance among various properties such as thermal stability, mechanical properties, and flame retardancy is very important for high-performance flame-retardant TPEE. Tuning the performance of final materials through tailoring the structure of metal salts of the phosphinate has been carried out in recent years. A series of aluminum salts, such as aluminum hypophosphite [23], aluminum isobutylphosphinate [24], aluminum phenylphosphinate [25], aluminum hydroxymethylphosphinate [26], etc., have been comprehensively investigated.

In our previous investigation, aluminum beta-carboxylethylmethylphosphinate, multi-arm aluminum phosphinates, and amide-containing phosphinate salts with varying metal cation and organic groups as flame retardants for epoxy, polybutylene terephthalate and TPEE have been reported [22,27,28,29]. It was found that the aluminum phosphinate with a longer alkyl group showed better compatibility and fewer negative effects on the mechanical properties of EP and polybutylene terephthalate.

To achieve a required flame-retardant rank while maintaining the mechanical properties at a satisfying stand, adding a synergistic agent to reduce the dosage of the flame retardant is a practical approach. For instance, Wang prepared an aluminum diethyl hypophosphite intercalation-modified montmorillonite nano-size flame retardant (AlPi-MMT) for TPEE. It was found that TPEE with 15 wt% AlPi-MMT exhibited better char formation and flame-retardant properties compared to those incorporating 15 wt% of AlPi or MMT alone [30]. Compared to inorganic compounds, the organic compounds showed better compatibility with TPEE. Some benzene or hetero-ring-rich compounds such as novolac [31], Belta-cyclodextrin [32], and triazine-containing compounds [33] have high-char-forming ability and good compatibility with TPEE. This is because benzene is a key component of residual charring, and benzene-rich compounds can effectively improve the charring performance of composites, thereby enhancing flame retardancy. For instance, Wang synthesized the triazine-based hyper-branched charring agent (CDS) as a char agent for TPEE/aluminum diethlyphosphinate flame-resistant composites. The results showed that the CDS, which could inhibit the melt dropping, improved the fire retardancy and mechanical properties [34]. Wu’s group reported a triazine-boron flame retardant (CPB). The CPB can improve the overall residual carbon performance of the material. When only 20% CPB is added, the TPEE composite achieves a high LOI value of 30.2% and passes the UL 94 V-0 rating [35].

In this work, aluminum butylmethylphosphinate (AiBMP), which has a chemical structure different from aluminum diethylphosphinate, was applied in the TPEE. AiBMP is a highly efficient flame retardant for TPEE. With 15 wt% AiBMP added, TPEE passed the UL 94 V-0 ranking with the LOI, improving from 20% to 36%. Phenolphthalein (Ph), a benzene-rich compound, was chosen as the additive combing with AiBMP to enhance the mechanical properties of TPEE. The advantages of phenolphthalein were identified by comparing the properties of TPEE/AiBMP/Ph and TPEE/AiBMP at the fixed filler loading.

## 2. Experimental Section

### 2.1. Materials

Thermoplastic elastomer (TPEE, hardness: 55D, melt flow index: 8.6 g/10 min at flowing rate of 2.16 kg at 220 °C, density: 1.17 g/cm^3^) was provided by Haiso Plastics Co., Ltd. (Wuhan, China). Aluminum butylmethylphosphinate (AiBMP) with a purity above 98% was provided by Zhenghao Chemical Ltd. (Wuhan, China). Phenolphthalein (Ph) is a chemical reagent that was purchased from Guoyao Chemical Ltd. (Tianjin, China) and used as received.

### 2.2. Sample Preparation

TPEE pellets were melt-blended with additives using an XK-160 twin–screw internal mixture (Changzhou, Jiangsu, China) at 220 °C for 20 min with a screw speed of 120 rpm. The resulting mixture was then hot-pressed at 225 °C for 5 min under 10 MPa for a sheet of suitable thickness and size for further measurements. The neat TPEE used as a standard was treated in the same way. The composition of the formulations is shown in Table 1.

### 2.3. Measurements

FTIR was recorded on a Tensor 27 Bruker spectrometer (Bruker, Karlsruhe, German) with KBr powder.

Thermogravimetric analysis (TG) was carried out with a TSDT Q600 spectrometer (TA, New Castle, DE, USA) thermogravimetric analyzer. A sample of about 10 mg was heated in an alumina pan from 30 °C to 700 °C at a linear heating rate of 20 °C/min in the nitrogen atmosphere.

Limiting oxygen index (LOI) measurements were taken using an HC-2-type instrum ent (Jiangning Analytical Instrument Factory, Nanjing, China) in accordance with ASTM D2863-97. The dimensions of the sample were 100 × 6.5 × 3 mm^3^, and five samples were carried out in the LOI test.

UL 94 vertical burning tests were conducted on a CZF-3 instrument (Jiangning Analytical Instrument Factory, Nanjing, China). The test was measured according to the vertical burning test standard ASTM D3801. The dimensions of the sample were 100 × 13 × 3 mm^3^, and three samples were carried out in the UL 94 test.

The heat release rate (HRR) and total heat release (THR) were measured in an MCC-2 microcombustion calorimeter (MCC) (Govmark Organization Inc., Farmingdale, NY, USA); samples of about 5–7 mg were heated in alumina pans from 40 °C to 700 °C at a heating rate of 1 °C/s. The flow rate of N_2_ and O_2_ was 80 mL/min and 20 mL/min, respectively.

Differential scanning calorimetry (DSC) measurements were carried out on a TA DSC-Q20 (Waltham, MA, USA) at a heating rate of 10 °C/min in N_2._ The flow rate of N_2_ was 50 mL/min.

Scanning electron microscopy (SEM) measurements were conducted using a Philips XL-40 instrument with a voltage of 15 kV. The sample was adhibitted on the copper plate.

X-ray diffraction (XRD) measurements were performed with an X-ray diffractometer (X’Pert Powder PANalytical, Almelo, The Netherlands) with CuKa radiation (1.5418 Å) at a scanning rate (2θ) of 5°/min. An X-ray fluorescent spectroscopy (XRF) measurement was conducted with a ZSX Primus II (Rigaku, Tokyo, Japan) XRF spectrometer with a 35 kV Rhanode tube.

The tensile strength and elongation of all specimens were measured by a Reger mechanical instrument (SUNS Company, Shenzhen, China) at a speed of 200 mm min^−1^ according to ASTM D412 standards, and five samples were conducted in the tensile test.

## 3. Results and Discussion

### 3.1. Characterization of AiBMP

Figure 1a presents the XRD spectrum of AiBMP. There are peaks at delta 8.1°, 14.0°, 21.5° and 28°, indicating how AiBMP exhibits a crystal structure. The composition of AiBMP was further analyzed with XRF, as shown in Figure 1b. Al is 21.72 wt%, and P is 72.28 wt%; the atomic ratio of P to Al is 3.13:1 (the calculated value is 3:1).

The SEM images of as-obtained AiBMP are shown in Figure 1c. AiBMP particles have an oval rod-like morphology with a length of about 50 μm, and rod-like AiBMP is composed of layers. The FTIR spectrum of AiBMP is shown in Figure 1d. The absorption peaks of about 3000–2800 cm^−1^ and 1462 cm^−1^, 803 cm^−1^, and 758 cm^−1^ are attributed to CH_3_ and CH2. The peak at 1288 cm^−1^ is for P-CH_3_. Other absorption peaks are 1150 cm^−1^ (P=O), 1080 cm^−1^ (P-O), and 882 cm^−1^ (Al-O) [23].

### 3.2. Combustion Characteristics: LOI, UL94, and MCC

The flame retardant properties of TPEE composites are studied using the LOI and UL94 vertical burning test, the results are presented in Table 1. TPEE is easily ignited, releasing smoke and serious flammable dripping The AiBMP is a very efficient flame retardant for TPEE. Adding 10 wt% AiBMP allows the LOI value of TPEE to increase from 20% to 29%. The sample TPEE-10 AiBMP was extinguished within 10 s after the first and second ignition. As the AiBMP content increased to 15 wt%, the LOI of TPEE-15AiBMP reached 36% with the V-0 rating achieved, and no dripping occurred during burning. Adding phenolphthalein has little influence on the fire-resistant properties of TPEE/AiBMP at low dosage (2 wt%). The LOI value of TPEE/TPEE-13AiBMP-2Ph is 35%, and the sample passed the V-0 rating. As the dosage of phenolphthalein increased to 4 wt%, LOI values of the samples TPEE-11AiBMP-4Ph reduced to 33%. The melt dripping was observed after the second ignition.

Figure 2 shows the heat release rate variation in samples with temperatures investigated with MCC. Data such as the peak of the heat release rate (PHRR), the temperature of PHRR (T_PHRR_), and the total heat release (THR) are presented in Table 2. Adding AiBMP reduces the PHRR and THR of the TPEE greatly. For instance, the PHRR and THR values of TPEE-15AiBMP were reduced by 55.5% and 14.57%, respectively, compared to the TPEE. In addition, the T_PHRR_ of TPEE containing AiBMP shifted slightly to a lower temperature due to the catalytic effect of AiBMP. When the total filler loading was kept at 15 wt%, the AiBMP was partially replaced with phenolphthalein; the PHRR, THR and T_PHRR_ of TPEE-13AiBMP-2Ph was close to that of TPEE-15AiBMP, while the THR and PHRR of TPEE-11AiBMP-4Ph increased to 23.9 kJ/g and 591.9 W/g, respectively. The above results indicate that phenolphthalein shows a negative effect on the heat release of TPEE at a high dosage (4 wt%).

### 3.3. Thermal Decomposition Behaviors

Figure 3 presents the TG-DTG curves of TPEE, phenolphthalein, AiBMP, and TPEE composites in N_2_. Detailed data, including temperature at 5% weight loss (T_5%_), the maximum rate degradation temperature (T_max_), the maximum decomposition rate (DTGmax), and char yields at 700 °C are summarized in Table 3. AiBMP is a highly thermal stable compound with T_5%_ 418.1 °C, T_max_ 486.1 °C, and residues of about 27.6% at 700 °C. Phenolphthalein decomposes very slowly. The residues of 35.4% are retained at 700 °C, demonstrating that phenolphthalein is a good char-forming agent. TPEE starts to lose weight at 373.2 °C with T_max_ 407.1 °C. It decomposes very fast and loses almost all of its weight before 420 °C.

With AiBMP added, the T_5%_ shifts to a lower temperature because of the catalytic effect of aluminum cation. The TPEE contains the ester group. It is well known that the thermal degradation of the polyester is easier when processed by the acid catalyst. AiBMP is a Lewis acid; it can accelerate the degradation of the TPEE in the early stage. The catalytic effect of the metal/organic complex has been reported in the literature [18,19,22,23]. However, the addition of AiBMP reduces the decomposition rate and improves the char yield of the TPEE. For instance, the DTG_max_ of TPEE is 2.58%/°C with a char yield of 4.3% at 700 °C. The DTG_max_ of TPEE-10AiBMP is 1.93%/°C, and the char yield is 5.94% at 700 °C. In the case of TPEE-15AiBMP, the DTG_max_ further reduced to 1.90%, and a char yield as high as 10.65 was achieved. Incorporating phenolphthalein increases the T_5%_ of the composites. The reason for this is that the phenol group of phenolphthalein as inhibitors decrease the activity of the free radical. Subsequently, the decomposition of TPEE/AiBMP/Ph moves to a higher temperature. In addition, the char residues of TPEE-13AiBMP-2Ph are higher than TPEE-15AiBMP. However, the DTG_max_ of TPEE-11AiBMP-4Ph increases, and the char yield is almost unchanged, with the dosage of phenolphthalein increasing to 4 wt%.

### 3.4. Residue Characterization

Figure 4 presents the digital and SEM photographs of the residues of TPEE (a), TPEE-15AiBMP (Figure 4b), and TPEE-13AiBMP-2Ph (c) collected by heating samples at 500 °C for 3 min in a muffle oven under a nitrogen atmosphere. The residues of TPEE are loose and composed of tiny ashes. The residues of TPEE-15AiBMP are smoother and condense with cracks and holes on the surface. SEM photo shows holes of about 3–8 um in size. The holes that arise are volatile during combustion. In addition, lots of micro-spheres are seen in the residues of TPEE-15AiBMP under SEM. The micro-spheres are formed by molten TPEE covering AiBMP residues. The micro-spheres are a barrier to hinders TPEE’s flow. The surface of TPEE-13AiBMP-2Ph residues is very condensed, smooth, and brilliantly black. A few tiny micro-spheres are found on the surface under SEM. The results from Figure 4 show that AiBMP acts in both the gas and condensed phases. The phenolphthalein acts mainly in the condensed phase by constituting a continuous barrier to prevent or slow down the diffusion of molten TPEE and volatile.

Figure 5 shows the heating and later cooling of the samples recorded using DSC. The crystallization temperature (Tc) and enthalpy (ΔHc) during cooling are listed in Table 4. The Tc of TPEE is 145 °C, and the Tc of TPEE/AiBMP composites moves to around 155 °C. The reason for this is that the AiBMP particles function as a nucleation agent to inhibit super-cooling; later, the chain-folding occurs at a higher temperature.

The ΔHc value is the degree of crystallinity. The higher ΔHc value indicates increased crystallinity. The ΔHc value of TPEE is 82.8 J/g. The ΔHc values of TPEE-10AiBMP and PEE-15AiBMP are 87.7 J/g and 50.2 J/g, respectively. The reason for this is that AiBMP particles act as a nucleation agent at low dosage (10 wt%) and promote the crystallization of TPEE, while the AiMBP particles dilute the local concentration of the TPEE at a high dosage (15 wt%) and retard the chain folding. As phenolphthalein was added, the huge benzene group of phenolphthalein, as well as hydrogen bonding between the phenolphthalein and TPEE, constrained the chain movement and results of the degree of decreasing crystallinity. Subsequently, the ΔHc values of TPEE-13AiBMP-2Ph and TPEE-11AiBMP-4Ph reduced to 9.6 J/g and 10.3 J/g.

### 3.5. Mechanical Properties

The tensile properties of TPEE composites are shown in Figure 6, The tensile strength and elongation at breaking both decrease with increasing the loading of AiMBP and then increase with the addition of phenolphthalein. For instance, the tensile strength and elongation at break of TPEE-15AiBMP were reduced by 53.0% and 42.8%, respectively, compared to the TPEE. Similar results were reported in a previous study. The reason corresponds to the poor compatibility between TPEE and AiBMP, leading to discontinuities at the particle/matrix interface.

Adding phenolphthalein improved the mechanical properties of composites. For instance, the tensile strength and elongation at break TPEE/13AiBMP/2Ph composites increased by 29.63% and 4.8%, respectively, in comparison to TPEE-15AiBMP. The enhancement in tensile properties resulted from the improved compatibility of phenolphthalein and the decreased mass fraction of AiBMP.

Figure 7 presents the SEM photos of the fractured cross-section of the samples after the tensile test. The AiBMP rods are not found in the matrix because they are wrapped by the matrix. There are lots of white irregular wrinkles on the surface of TPEE-15AiBMP. Many dimples are seen on the surface of TPEE-13AiBMP-2Ph. Generally, breaking often occurs at the interface of the hard filler and the polymer matrix. These wrinkles are caused by the dilute deformation of the polymer near the AiBMP surface. Proportionally, substantial amounts of energy are needed to induce dilute deformation. The dimples are denser and smaller on the surface of TPEE-13AiBMP-2Ph than that of TPEE-15AiBMP, showing how phenolphthalein enhances the interface adhesion between AiBMP and TPEE.

## 4. Conclusions

In this study, a novel aluminum salt of organic phosphinate AiBMP as the main flame retardant and benzine-rich compound phenolphthalein as the synergistic agent were applied as the flame retardants of TPEE. The results found that AiBMP alone is an efficient flame retardant of TPEE. The TPEE containing 15 wt% AiBMP passed the UL 94 V-0 rating and achieved a high LOI value of 30.2%. TG analysis showed how the addition of AiBMP reduced the decomposition rate and improved the char yield of the TPEE. Results from the SEM of residues indicate that AiBMP acts in both gas and condensed phases.

Phenolphthalein exhibits an impressive enhancement in mechanical properties of TPEE. When the total filler loading is kept at 15 wt%, the tensile strength and elongation of the sample TPEE-13AiBMP-2Ph increased by 29.63% and 4.8%, respectively, in comparison to the TPEE-15AiBMP and flame-retardant properties of TPEE. The plasticizing effect of phenolphthalein and the uniform dispersion of AiBMP are the main reasons for the improved tensile strength and high retention of the elongation at the breaking of TPEE-13AiBMP-2Ph.

Moreover, the benzene-rich structure of phenolphthalein endows good char formation and enhances flame retardancy in the solid phase. TG results showed that the char yield of TPEE-13AiBMP-2Ph at 700 °C improved by 0.5% compared to that of TPEE-15AiBMP. SEM morphology indicated that the surface of TPEE-13AiBMP-2Ph residues is more condensed and smoother than that of TPEE-15AiBMP.

With the development and popularization of electric vehicles, corresponding charging facilities with high insulation and safety are required. The AiBMP and phenolphthalein-based flame retardant TPEE have potential applications as a charging cable and pile. In future work, we aim to evaluate the hydrolysis resistance and electrical properties of TPEE/AiBMP/Ph composites under a high-voltage environment.

## Figures and Tables

**Figure 1 polymers-16-00552-f001:**
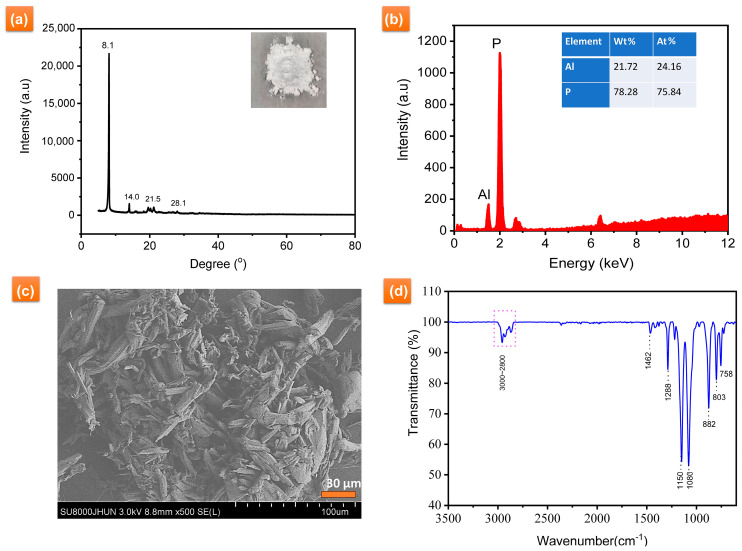
(**a**) XRD spectra, (**b**)XRF, (**c**) morphology and (**d**) FTIR spectra of AiBMP.

**Figure 2 polymers-16-00552-f002:**
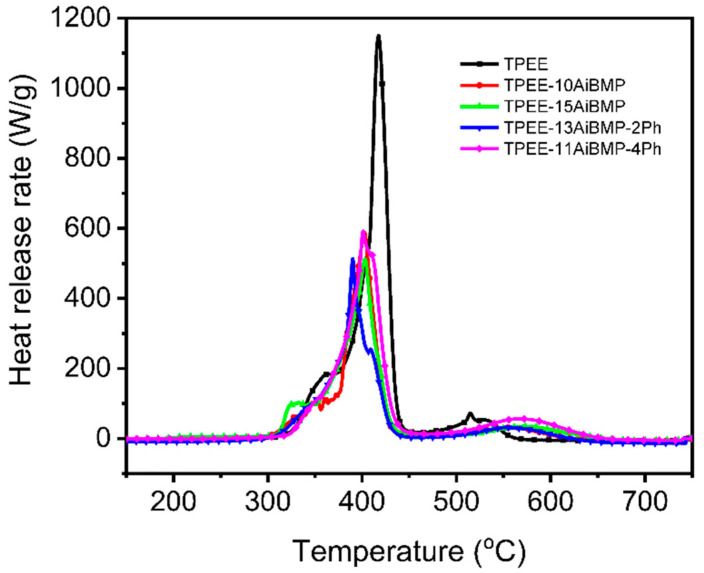
Heat release rate vs. temperature of samples.

**Figure 3 polymers-16-00552-f003:**
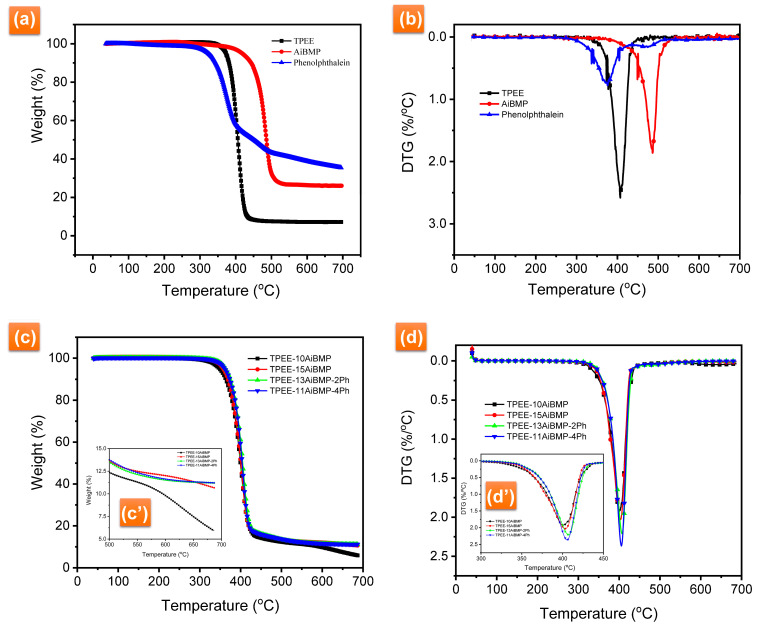
(**a**) TG and (**b**) DTG curves of TPEE, phenolphthalein and AiBMP in N_2_. (**c**) TG and (**d**) DTG curves of TPEE composites in N_2_. (**c′**) TG curves of TPEE composites between 500–700 °C in N_2_, (**d′**) DTG curves of TPEE composites between 300–450 °C in N_2_.

**Figure 4 polymers-16-00552-f004:**
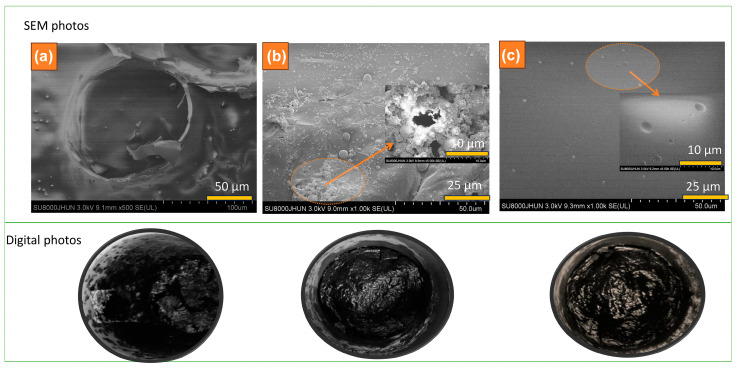
SEM (upper) and digital photos (bottom) of the residues TPEE (**a**), TPEE-15AiBMP (**b**) and TPEE-13AiBMP-2Ph (**c**) obtained by heating samples at 500 °C for 5 min.

**Figure 5 polymers-16-00552-f005:**
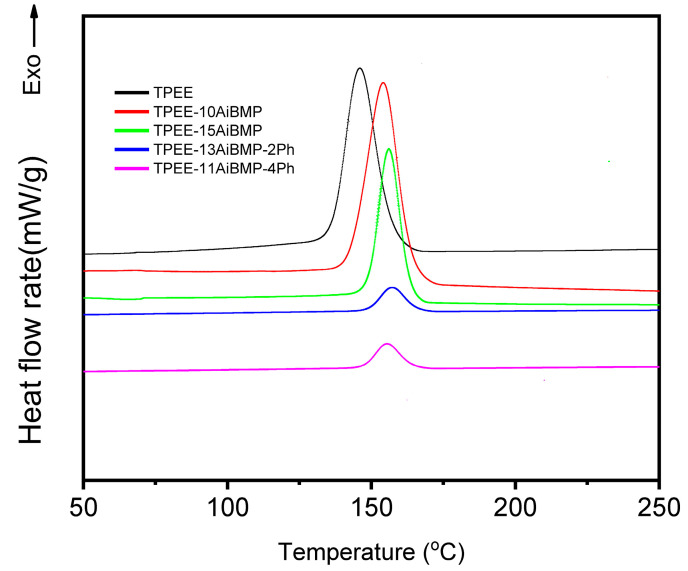
DSC curves of heating and subsequent cooling for samples.

**Figure 6 polymers-16-00552-f006:**
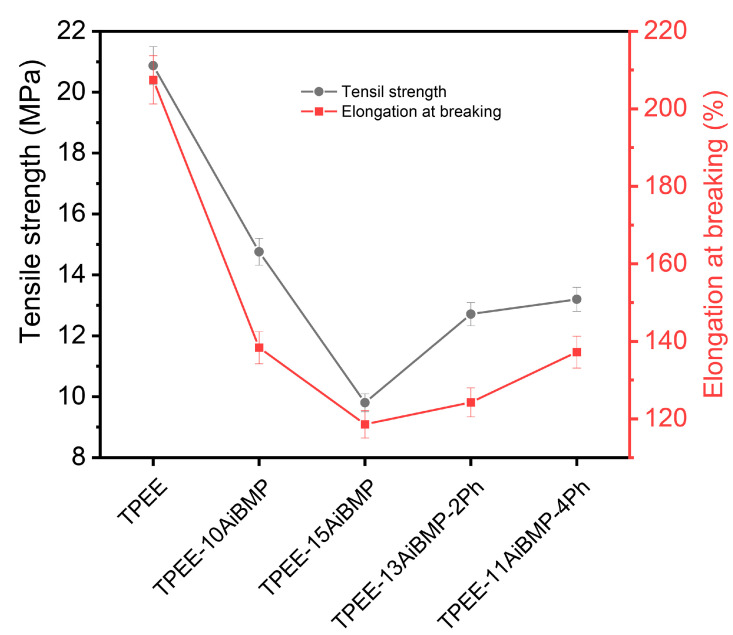
Tensile properties of samples at a crosshead speed of 25 mm/min.

**Figure 7 polymers-16-00552-f007:**
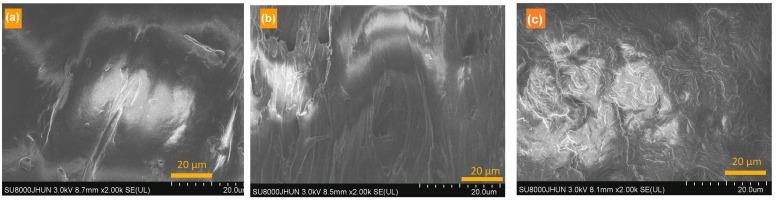
SEM images of fractured surface for TPEE (**a**), TPEE-15AiBMP (**b**) and TPEE-13AiBMP-2Ph (**c**).

**Table 1 polymers-16-00552-t001:** Formula of the flame-retardant TPEE.

Sample	TPEE (wt%)	AilBMP (wt%)	Ph (wt%)	LOI (%)	UL94 Rating	Dripping
TPEE	100	0	0	20	-	yes
TPEE-10AiBMP	90	10	0	29	V-1	yes
TPEE-15AiBMP	85	15	0	36	V-0	no
TPEE-13AiBMP-2Ph	85	13	2	35	V-0	no
TPEE-11AiBMP-4Ph	85	11	4	33	V-1	yes

**Table 2 polymers-16-00552-t002:** MCC data of the samples.

Sample	PHRR (W/g)	THR (kJ/g)	T_PHRR_ (°C)
TPEE	1150.4	24.7	417.7
TPEE-10AiBMP	588.1	22.3	402.7
TPEE-15AiBMP	511.9	21.2	403.4
TPEE-13AiBMP-2Ph	513.6	20.9	390.1
TPEE-11AiBMP-4Ph	591.9	23.8	407.4

**Table 3 polymers-16-00552-t003:** TG-DTG data of the samples in N_2_.

Sample	T_5%_ (°C)	T_max_ (°C)	DTG_max_ (%/°C)	Residues at 700 °C (%)
TPEE	373.2	407.1	2.58	4.30
AiBMP	418.1	486.1	1.86	26.0
phenolphthalein	323.4	375.8	0.78	35.4
TPEE-10AiBMP	348.7	400.3	1.93	5.91
TPEE-15AiBMP	357.8	402.8	1.90	10.7
TPEE-13AiBMP-2Ph	363.1	407.4	2.21	11.2
TPEE-11AiBMP-4Ph	361.2	405.5	2.37	11.2

**Table 4 polymers-16-00552-t004:** Data from DSC curves of the samples.

Sample	Tc (°C)	ΔHc (J/g)
TPEE	145.0	82.2
TPEE-10AiBMP	154.0	87.7
TPEE-15AiBMP	155.8	50.2
TPEE-13AiBMP-2Ph	155.6	9.1
TPEE-11AiBMP-4Ph	155.1	10.3

## Data Availability

Data are contained within the article.

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
