# Peer review of "Flame-Retardant Thermoplastic Polyether Ester/Aluminum Butylmethylphosphinate/Phenolphthalein Composites with Enhanced Mechanical Properties and Antidripping"

_polymers, 2024, doi:10.3390/polym16040552_

Round 1

Reviewer 1 Report

Comments and Suggestions for Authors

The topic of flammability of plastics is still current and up-to-date. The development of new flame retardants and research methods will make it possible to limit the flammability of polymers and thus enable their use in many industries due to the appropriate regulations regarding fire safety.

The introduction to the manuscript is well prepared, but it is worth pointing out the potential use of such a flame-retardant material. The literature review is appropriate. The cited literature provides a good background to the problem. Literatu is a bit modest, but up-to-date. The authors did not avoid self-citations.

The work methodology is presented correctly, it might be worth describing the research methods in detail and checking whether the standards used are up to date

Figures and tables are correctly described. The descriptions in the main text are also supplemented by comments to tables and figures.

The results obtained are quite interesting, especially the combined LOI and UL-94. Was the smoke resistance of the obtained materials specifically determined?

In the case of final conclusions, it is worth extending them with a recommendation regarding the use of such flame-retardant material. It is worth highlighting three main conclusions and what further work will look like.

Author Response

Dear reviewer,

Thank you for your kindness and constructive comments. We benefit a lot from these comments. It not only helps us to improve the quality of the manuscript, but also enhances our research ability in the future study.

  We have finished the amendment of manuscript and responded, point by point to the comments. The paper has been revised throughout the text.

Reply to Reviewer 1

Comments and Suggestions for Authors

The topic of flammability of plastics is still current and up-to-date. The development of new flame retardants and research methods will make it possible to limit the flammability of polymers and thus enable their use in many industries due to the appropriate regulations regarding fire safety.

The introduction to the manuscript is well prepared, but it is worth pointing out the potential use of such a flame-retardant material. The literature review is appropriate. The cited literature provides a good background to the problem. Literature is a bit modest, but up to date. The authors did not avoid self-citations.

Reply: The potential use of such a flame-retardant material has been pointed out in the introduction section (page 2/12, line 59-62) and the conclusion section (page10/12,line 309-313). The introduction has been enriched with recent references.

The work methodology is presented correctly, it might be worth describing the research methods in detail and checking whether the standards used are up to date.

Reply: We have checked the research method and described it more detailed. Revised part has been marked in blue, as shown in the page 3/12 and 4/12 (2.3. Measurements)

Figures and tables are correctly described. The descriptions in the main text are also supplemented by comments to tables and figures.

Reply: Thanks for the good suggestion. We have rich the descriptions in the main text to the tables and figures.

The results obtained are quite interesting, especially the combined LOI and UL-94. Was the smoke resistance of the obtained materials specifically determined?

Reply: Thank you for your comments. Smoke resistance of composites was not determined in current study. We would like to do it in the coming work.

In the case of final conclusions, it is worth extending them with a recommendation regarding the use of such flame-retardant material. It is worth highlighting three main conclusions and what further work will look like.

Reply:  We have revised the conclusion and highlights are summarized, according to the comments. Please see the revised conclusion (page 12, line 309-313).

Reviewer 2 Report

Comments and Suggestions for Authors

1. The Introduction section should be enriched with recent references comparing the effectiveness of different types of flame retardants. Typing errors like “g/cm3, N2, mm min−1 etc.” can be avoided.

2. XRD, FTIR, and XPS are provided only for for procured AiBMP. These tests can be conducted for composite samples and the results could be included.

3. SEM micrograph of AiBMP alone is given. The morphological analysis of the composite sample will give a better understanding of mechanical and thermal properties.

4. Why does the sample TPEE-15AiBMP sample show better results in LOI, UL-94, and MCC? Explain the catalytic effect of AiBMP in detail.

5. Why does phenolphthalein negatively affect the heat release of the TPEE at high dosage (4 wt%)?

6. In “Figure 3. TG-DTG curves of samples under nitrogen atmosphere” a, b, c, and d are not specified

7. In Figure 3. (3rd fig ) blue line ( TPEE-11AiBMP-4Ph) has appreciable right shift than TPEE-13AiBMP-2Ph . However, in Table 3 the trend is different at T5%(oC). Explain this contradiction.

8. A table can be added in the DSC section mentioning the important parameters for easy reference

9. Why tensile strength of all the filler-incorporated samples is lesser than neat TPEE? Do you mean to say that the interphase is weak and the stress transfer is not effective.?

10. Tensile Stress-strain graphs for the samples should be provided and a table for tensile properties with standard deviation/ error bar should be included.

11. Provide very crisp statements in the Conlusion

Comments on the Quality of English Language

Typing errors like “g/cm3, N2, mm min−1 etc.” can be avoided. Also grammar should be checked through out

Author Response

Dear reviewer,

Thank you for your kindness and constructive comments. We benefit a lot from these comments. It not only helps us to improve the quality of the manuscript, but also enhances our research ability in the future study.

  We have finished the amendment of manuscript and responded, point by point to the comments. The paper has been revised throughout the text.

Sincerely

                                                    Xueqing

Reply to Reviewer 2

Comments and Suggestions for Authors

  1. The Introduction section should be enriched with recent references comparing the effectiveness of different types of flame retardants. Typing errors like “g/cm3, N2, mm min−1 etc.” can be avoided.

 Reply: We have enriched introduction with recent references [refer 8-10,30,35], as shown in the introduction section. In addition, typing errors have been corrected.

  1. XRD, FTIR, and XPS are provided only for procured AiBMP. These tests can be conducted for composite samples and the results could be included.

 Reply: AiBMP is a novel compound, therefore its structure is provided. The TPEE is a commercial compound, its structure has been published in literature. The results showed that the addition of AiBMP has no effect on the structure of TPEE due to AiBMP physically blending with TPEE. Considering no contribution to the results, the XRD, FTIR and XRF of composites were not included in the manuscript.

We would like to provide it to the reviewer or reader as a supplementary materials when necessary.

  1. SEM micrograph of AiBMP alone is given. The morphological analysis of the composite sample will give a better understanding of mechanical and thermal properties.

 ReplyThe SEM photos of the composite sample are presented in Figure 7. The morphological analysis of the composite sample has been revised accordingly (page 8/12.line 281).

  1. Why does the sample TPEE-15AiBMP sample show better results in LOI, UL-94, and MCC? Explain the catalytic effect of AiBMP in detail.

  Reply: The flame-retardant performance of the composites depends on the loading of the flame retardant. The AiBMP plays the key role in flame retardancy. With the loading of AiBMP increasing, the LOI and UL-94 rating of TPEE increases. The heat releasing rate and total heat releasing of the composites reduced.

Phenolphthalein is a synergistic agent. The functions of phenolphthalein are promoting the char residues in the condensed phase, and as a plasticizer improve compatibility between the AiBMP and TPEE.

  When the phenolphthalein partially replaces the AiBMP, the flame retardancy of the composites changed a little.

We have added explanation to the catalytic effect of phenolphthalein in detail, as shown in the manuscript (Page 7/12, Line 206-210). TPEE contains the ester group. It is well known that thermal degradation of polyester is easily procced by acid catalyst. AiBMP is a Lewis acid. Therefore, it can accelerate the degradation of the TPEE. The catalytic effect of metal/organic complex has been reported in literatures as shown in revised manuscript (Rer.18,19,22,23).

  1. Why does phenolphthalein negatively affect the heat release of the TPEE at high dosage (4 wt%)?

  Reply: Thank you for your question. We have measured samples with MCC several times, and the results showed that the heat release of TPEE-11AiBMP-4Ph is higher than the others. The probability reason is the plasticizing effect of the phenolphthalein, which dilutes the TPEE and leads to the spreading of heat faster. In addition, the loading of AiBMP reduced in the TPEE-11AiBMP-4Ph, which could not depress the heat release of TPEE efficiently.

  1. In “Figure 3. TG-DTG curves of samples under nitrogen atmosphere” a, b, c, and d are not specified

 Reply: The a, b,c,d have been specified in the TG-DTG cures, as shown in the corrected Figure 3.

  1. In Figure 3. (3rd fig ) blue line ( TPEE-11AiBMP-4Ph) has appreciable right shift than TPEE-13AiBMP-2Ph . However, in Table 3 the trend is different at T5%(oC). Explain this contradiction.

Reply: Thanks for your careful observing and comments. We have checked Figure 3 and the original data. The results show that the T5% is 361.2 oC for TPEE-11AiBMP-4Ph and 363.1 oC for TPEE-11AiBMP-4Ph.

 We guess that the direction of triangle symbols in the curve of TPEE-11AiBMP-4Ph causes the visual errors.

  1. A table can be added in the DSC section mentioning the important parameters for easy reference

ReplyThe table has been added, as shown in Table 3 (page 7/11). In addition, the mark in Figure 5 is removed.

  1. Why tensile strength of all the filler-incorporated samples is less than neat TPEE? Do you mean to say that the interphase is weak, and the stress transfer is not effective.?

Reply: Yes, the reduction in the tensile strength of composites is due to the weak interphase.

  1. Tensile Stress-strain graphs for the samples should be provided and a table for tensile properties with standard deviation/ error bar should be included.

ReplyThe results of Figure 6 are based on the average value of five samples. There are five stress-strain curves for each formula. Considering a single stress-strain curve could not present the tensile properties shown in Figure 6, we did not add the Tensile Stress-strain graph in the revised manuscript. We would like to provide the original data of the mechanical test when necessary.

 According to the comment of the reviewer, we have added the error bar to the Figure 6.

  1. Provide very crisp statements in the Conclusion

 ReplyConclusion has been rewritten, as shown in the revised Manuscript.

Comments on the Quality of English Language

Typing errors like “g/cm3, N2, mm min−1 etc.” can be avoided. Also grammar should be checked through out

 Reply: Typing and grammar have been checked and errors were corrected.

Round 2

Reviewer 2 Report

Comments and Suggestions for Authors

The modifications made are satisfactory